# Role of the Androgen Receptor in Gender-Related Cancers

**Emilia Sabbatino** †, **Viviana Tutino** †, **Fabrizio Licitra**, **Marzia Di Donato** [ID], **Gabriella Castoria** [ID], **Antimo Migliaccio** [ID] **and Pia Giovannelli** *[ID]

Department of Precision Medicine, University of Campania "L. Vanvitelli", Via L. De Crecchio 7, 80138 Naples, Italy
* Correspondence: pia.giovannelli@unicampania.it
† These authors contributed equally to this work.

**Abstract:** The androgen receptor (AR) is expressed in many cell types, and its related signaling is widely investigated in hormone-dependent cancers such as prostate and breast. The significance of the AR, however, has been detected even in other cancers, including gastric, bladder, kidney, lung, hepatic, and pancreatic, in which growth and spreading are not strictly or notoriously dependent on sex steroid hormone action. The incidence and mortality of these cancers are, however, somewhat related to gender and, specifically, are higher in men than in women, with the ratio reaching 3–4:1 for bladder cancer. This direct correlation between cancer incidence, mortality, and gender makes sex one of the most important risk factors for these cancers and has incited investigation about the role of sex steroid receptors and their activating hormones in gender-related cancers. In these cancers, the AR is often expressed and seems to play a pivotal role in different processes contributing to cancer onset and progression such as growth, spreading, and epithelial to mesenchymal transition (EMT). This manuscript will offer an overview of the role of the AR in many cancers of the respiratory and gastrointestinal systems wherein its role has been at least partially analyzed. Understanding the role of the AR in these tumors could help us to identify a new biomarker for early diagnostic guidance and to develop better therapeutic approaches by directly targeting the AR or its downstream signaling in individual cells of hormone-related cancers at different stages.

**Keywords:** androgen receptor; gender-related cancers; sex steroid hormones in cancers

## 1. Introduction

Cancer ranks among the most common causes of death worldwide, and its incidence and mortality are expected to increase due to both the aging of the population and the major diffusion and worsening of some of the risk factors responsible for its onset, such as pollution or an unhealthy lifestyle. In addition to the classic environmental and genomic risk factors, the incidence and mortality of a lot of cancers are also determined by sex or gender. For this reason, an increasing number of scientists have been studying the role of sex steroid hormones in many cancers in addition to reproductive cancers, in which the hormone/hormone receptor action is absolutely the principal guide [1]. To date, different cancers have shown gender disparities, not only in incidence but also in aggressiveness and disease prognosis. Except for breast, thyroid, and other rare cancers located in specific sites of the digestive system, a lot of cancers, such as lung, kidney, bladder, gastric, colorectal, liver, and pancreatic, as well as hepatocarcinoma and many others, show a higher incidence in males [2]. The mechanisms underlying this phenomenon are completely unknown, but there are some clear leading points that can help to understand these cancer-related gender disparities. Occupational risk factors, differences in levels of circulating hormones, and the expressions of their receptors could represent starting points to explain gender disparities in patients with cancers with a higher incidence in males [3]. Even if, between the two sexes, there are no differences in the pivotal mutated genes participating in a cancer's development, as is the case with the BRAF gene in melanoma or K-RAS in pancreatic

cancer, we must consider that there are whole groups of genes differentially expressed in response to sex steroid hormones able to influence several processes in cancer. For example, studies analyzing gene expression in clear cell renal carcinoma (ccRCC) have shown that, among the analyzed genes, about 90% were activated in a gender-specific way [3]. Accumulating evidence displays that gender differences also influence the immune system, thereby contributing to the unequal disease outcomes and different efficiency in immune response to therapies in men and women [4].

By reason of the higher incidence in the male gender of many cancers, it is suitable to have a better understanding of the role of sex steroid receptors in gender-related cancers; in particular, it could be advantageous to analyze the role of the androgen receptor (AR) in these gender-related cancers.

AR action has been extensively studied in hormone-dependent cancers such as prostate and breast.

In addition to the classical pathway, the AR can activate the non-classical, or rapid, pathway by an alternative mechanism mediated by different signaling proteins [5,6].

In prostate cancer (PCa), the AR represents a key regulator of tumor development and progression. Several studies in PCa patients treated with anti-androgen or androgen ablation therapy have revealed how androgen/AR signaling mediates many physiological and pathophysiological processes in various tissues/organs [7–10].

In AR-positive breast cancer, the role of the AR in cell proliferation, apoptosis, migration, and cell invasion is known [5]. Moreover, in triple-negative breast cancer (TNBC), a growing number of studies has clarified the mechanisms used by this receptor to promote cancer progression and aggressiveness [11,12].

In recent years, many researchers have devoted their attention to the actions of the AR in all those cancers not "classically" hormone-dependent, but gender-related, such as lung, kidney, bladder, liver, stomach, and pancreas.

In the group of "classically hormone-dependent cancers" are included all those cancers in which growth and invasiveness are notoriously and directly controlled by sex steroid hormones and their receptors, such as breast cancer in women and prostate cancer in men. Other examples are testicular cancer in men, and uterine and ovarian cancers in women. Except for breast cancer, occurring in both sexes with a clear predominance in women, all these cancers are also sex specific.

Conversely to these, the "non-classically hormone-related cancers", also known as gender-related cancers, include all those cancers occurring in both sexes, for which it is unknown if any dependence from sex steroid hormones and receptors exists, but there is still an incidence imbalance between men and women. This gap could be explained by the unequal concentrations of circulating hormones between men and women.

This review aims to collect and discuss data about the role of the AR in several gender-related cancers whose incidence and mortality are higher in men than in women. This receptor could be an attractive therapeutic target in tumors as well, and the use of antagonists, agonists, and modulators could be an alternative pharmacological strategy.

## 2. Androgen Receptor (AR)

The androgen receptor belongs to the large family of type I nuclear receptors. As such, it is a ligand-dependent transcription factor commonly activated by ligand binding. The AR gene is located on the long arm of the X chromosome (Xq11-12) and consists of eight exons that code for the three functional domains typical of steroid hormone receptors: (1) an amino-terminal domain (N-terminal domain, NTD), also indicated as a trans-activation domain (TAD, residues 1–555), (2) a DNA binding domain (DBD, residues 555–623), and (3) a carboxyl-terminal ligand binding domain (LBD, residues 665–919). Finally, a hinge region (residues 623–665) connects the DBD and the LBD [13].

The NTD or TAD represents a variable domain, less conserved than the others. It contains an activation region called AF-1 (ligand-independent transactivation domain) whose absence results in a transcriptional impairment of the receptor's functions. This

region is structurally flexible and is critical in stabilizing the receptor by enhancing the interactions with AR co-activators. The DBD is a highly conserved domain in nuclear receptors; it contributes to androgen receptor dimerization as well as to the binding of specific sequences in chromatin known as androgen response elements (AREs). The LBD turns out to be important in the nuclear localization of the AR. In this domain, there is a region termed AF-2, a ligand-dependent activation region, responsible for the complete activation of the receptor [10].

AR has two isoforms: AR-A and AR-B (Figure 1).

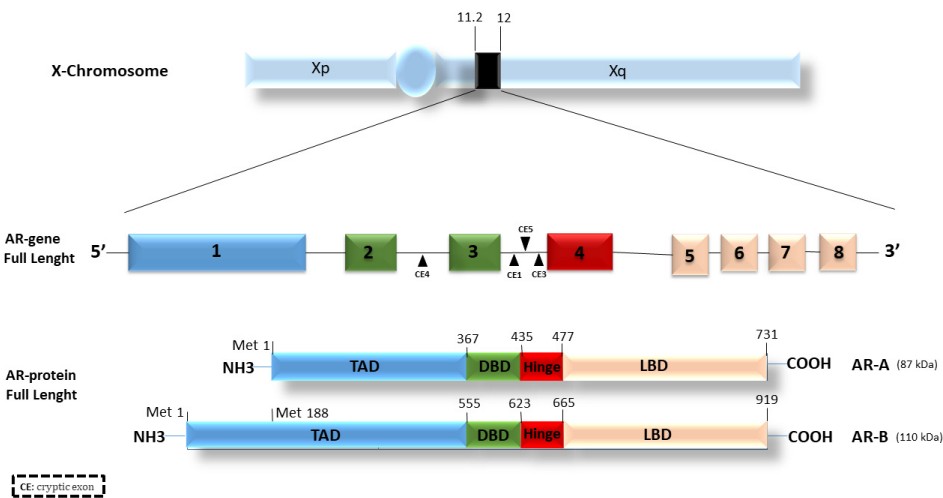

**Figure 1.** Androgen receptor structure and isoforms. The AR gene is located on the X chromosome and consists of 8 different exons encoding for three distinct functional regions: the TAD (transactivation domain), the DBD (DNA binding domain), and the LBD (ligand binding domain). The DBD and the LBD are linked by a hinge region. Different cryptic exons (CE) are located between exons, i.e., between exons 2, 3, and 4.

The isoform B, also named AR full-length or isoform 1, is widely expressed in most cell types, migrating with an apparent mass of 110 kDa. The transcription of the isoform A starts from the methionine in position 188 (Met-188). The resulting protein migrates with an apparent mass of 87 kDa, lacking 187aa in the N-terminal transactivation domain [14,15]. Both the isoforms are expressed in a sweeping variety of adult and fetal reproductive and non-reproductive tissues [16]. The full-length AR-B represents the predominant AR species in all tissues in which both isoforms have been detected, and, in male and female adult reproductive tissues, AR-B is expressed at high concentrations, whereas AR-A comprises 20% or less of the total AR protein [16]. Furthermore, the ratio of AR-A to AR-B was not shown to change widely in the tissues examined [16]. The two AR isoforms slightly differ in their activity and similarly respond to a variety of androgen agonists and antagonists [17]. The unique differences were studied by Liegibel and colleagues [18]. They proved that AR isoforms have distinct functions in human cells of mesenchymal origin such as osteoblastic cells and genital skin fibroblasts. AR-B was responsible for the mitogenic stimulation of mesenchymal cells, whereas, in AR-positive tissues, AR-A inhibited the mitogenic function of androgen-activated AR-B. AR-A was unable to stimulate cell proliferation, probably due to the reduced binding of AR co-activating protein to the truncated N-terminal TAD [18].

Both the isoforms are expressed in prostate cancer, wherein the AR-B level is still higher than AR-A. Anyway, the AR A/B ratio increases in PCa, in parallel with the Gleason score [15]. These results agree with those obtained in studies of colon cancer, wherein the AR-B expression decreased, whereas the AR-A expression was maintained [19]. The different results in AR-A and -B activity can be explained by considering that all the measurements of their activity were performed using similar levels of the two isoforms, but this does not replicate the normal conditions.

In addition to the classical isoforms, ARs frequently undergo mutations or alternative splicing that causes the formation of alternative splicing variants. Some of these different forms of ARs, represented in Figure 2, can be expressed in normal and cancer tissues and can trigger altered and uncontrolled responses, causing various pathologies or drug resistance in cancer, such as for AR-V3, -V7, or AR-8 in PCa [20–22] or for AR-V45 in BC [12]. According to the NCBI site (https://www.ncbi.nlm.nih.gov/gene/367#reference-sequences, accessed on 10 May 2023), there are 5 AR isoforms, indicated as AR 1, 2, 3, 4 and 5 and corresponding to AR-B, AR-V45, AR-V7, AR-V1 and AR-8, respectively.

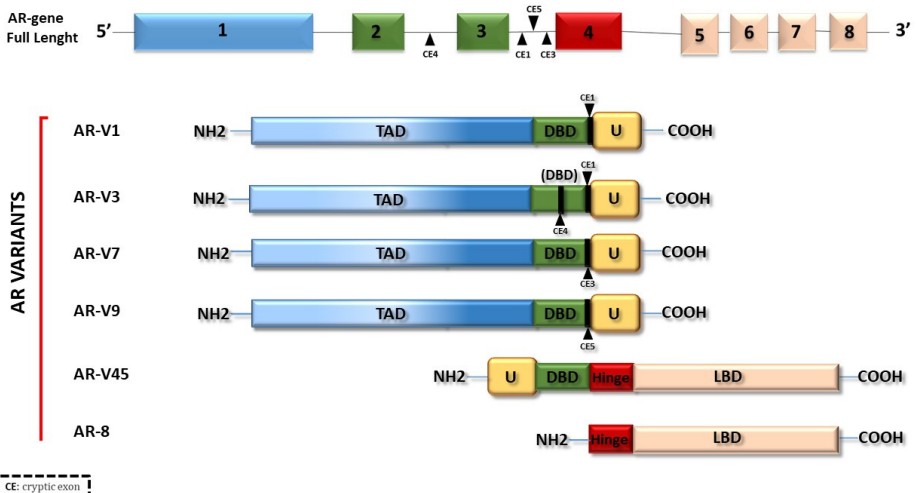

**Figure 2.** AR variants. The figure illustrates some of the naturally occurring AR variants. Most of them originate from AR alternative splicing.

The presence of the AR and/or its variants makes more complex the molecular scenario of different types of cancers. Studies on the alteration of agonist/antagonist properties of anti-androgens due to different AR mutations have stimulated the search for new drugs that inhibit AR signal transduction [23,24]. It has been seen how mutations in the AR gene and splicing variants can lead, in most cases, to increased cellular aggressiveness in hormone-dependent cancers such as prostate, breast, and ovary [12,25–31]. All the AR domains can undergo mutations. In particular, the selection pressure of drugs on the AR pathway in PCa increases the number of mutations in the ligand binding domain (LBD), thereby broadening its ligand specificity and sensitivity and reducing the clinical treatment effects of PCa and the quality of patient survival. Although various AR mutations have been reported in prostate cancer, specific hot spot mutations (L702H, W742L/C, H875Y, F877L, and T878A/S) were frequently identified after the gain of drug resistance [32]. A conspicuous group of AR splicing variants, such as AR-V1, 3, 7 and 9, lacks the LBD while showing an intact NTD and DBD and, consequently, a constitutive activity [33–35]. Other variants such as AR-V45 and AR-8 lack a DBD and do not work as transcription factors but play different roles. AR-8 promotes cell survival via a non-genomic mechanism [36,37]. AR-V45 is an NTD truncated form, unable to transactivate AR but able to work as a dominant negative and suppress the AR FL functions [37]. The hormone-activated AR works through a genomic or classical pathway and a non-genomic or rapid mechanism. In the genomic mechanism, the AR dimerizes after the binding to its hormone and binds AREs in the promoter region of target genes involved in cell proliferation, epithelial to mesenchymal transitions (EMT), apoptosis, and metabolism [38]. On the other side, the activated AR is also able to influence biological processes, triggering the activation of signaling pathways in few seconds or minutes. These two modes of action are not unconnected, and they work together in same target cells [39], thereby promoting the growth and development of hormone-dependent tumors [40].

The role of the AR has been widely studied in prostate cancer. The AR controls cancer progression and its action is inhibited by reducing androgen synthesis or using specific

AR antagonists. However, resistance to these treatments often occurs within 2–3 years of therapy, when patients develop castration-resistant prostate cancer (CRPC), in which active ARs remain as key regulators. Some studies have focused on the functional domains of the AR and its crucial role in this pathology. Deeper knowledge of the structures of the DBD and LBD of the AR provides a framework for understanding the functions of this receptor and can lead to the design of drugs for the treatment of prostate cancer [10].

A great deal of scientific evidence shows the involvement of the AR in female cancers such as ovarian and breast.

Although the molecular mechanisms underlying the androgen receptor role in ovarian cancer are still far from being fully understood, therapeutic approaches designed to modulate androgen receptor activity induce a decrease in tumor progression [27].

In breast cancer (BC), the AR seems to play opposite roles, and this could be attributed to its crosstalk with other signaling pathways. Anyway, whereas in estrogen receptor (ER)-positive breast cancer, the AR appears to both inhibit and promote tumor progression [5,41], in ER-negative BC, particularly in triple-negative breast cancer (TNBC), it mainly promotes tumor progression and tumor growth [5].

It is of paramount importance to increase our understanding of whether the AR signaling pathway also influences the oncogenesis, or the growth and spreading, of gender-related tumors, in order to use targeted strategies to slow tumor progression.

## 3. AR in Lung Cancer

Lung cancer is the leading cause of cancer death in the world (http://globocan.iarc.fr/, accessed on 10 February 2023). The high mortality is due to late diagnosis, because of the paucity of symptoms in the early stages of this cancer.

Risk factors commonly associated with lung cancer are, above all, tobacco consumption, but also include occupational exposure to cancer-causing agents such as asbestos, radon, and air pollution [42].

Furthermore, the disease shows sex and gender differences, with a higher incidence in men than women [43] and with sex ratios of men to women varying from 1.5 to 2.0 to 1 (http://globocan.iarc.fr, accessed on 10 February 2023). Females tend to be diagnosed younger, at earlier stages and, mostly, with a better prognosis [44]. Worldwide lung cancer mortality is around threefold higher in males, with a current downward trend for males and upward trend for females [44,45]. This trend will slightly change in 2023 in the United States, where the lung cancer incidence will be slightly higher in women than in men, inversely to mortality, which will remain higher in men [45]. Additionally, female patients show better survival rates than males at any stage of disease [46]. Overall, men tend to be less vulnerable to tobacco carcinogens than women [47]. These differences may be due to the increased expression of AR in lung cancer cells [16].

The scientific evidence shows that the adult lung is a target tissue for ARs and suggests that the AR plays a role in lung cancer biology. Mikkonen and colleagues observed not only that the AR is expressed in different human lung cancer types, but also that its expression increases, after androgen treatment, in murine lungs, above all in type II pneumocytes and the bronchial epithelium [48]. Furthermore, androgen treatment significantly alters the gene expression profile in the murine lung and in A549-lung-cancer-derived cells by upregulating transcripts involved in oxygen transport and downregulating those responsible for DNA repair and recombination [48].

Some studies highlight how AR signaling in lung cancer influences tumor progression. Recchia and colleagues demonstrated that in non-small cell lung cancer (NSCLC) cells, specifically in A549 cells, AR and EGFR co-work to trigger cell growth. The molecular mechanisms underlying the crosstalk between growth factors and steroid hormones have been studied, mainly in androgen-sensitive PCa LNCaP cells expressing both ARs and EGFRs [49]. The interaction between dihydrotestosterone (DHT) and ARs controls the expression of AR-responsive genes, such as cyclin D1, at both the transcript and protein

levels. In A549 cells, the cooperation between ARs and EGFRs activates p38, thereby regulating the CD1-mTOR pathway and lung cancer proliferation and progression [50].

Similar results were obtained by Lu and co-workers, who reported the downregulation of cyclin D1 and the suppression of cell proliferation and anchorage-independent growth after AR-knockdown via AR-siRNA in A549 cells. In these cells, the AR-siRNA reduced the expression level of another protein, OCT4, implicated in tumor progression and metastasis [51].

In addition to cyclin D1 and OCT4, there is a multitude of proteins whose transcription is AR-dependent. Angiopoietin-like 4 (ANGPTL4) is upregulated by androgens in murine lung and is involved in the selective formation of lung metastases under transforming growth factor (TGF) control [52]. TMPRSS2 gene results upregulated upon androgen exposure [48]; it is a target of ARs in the prostate [53] and a locus for translocation of erythroblast transformation specific (ETS) transcription factors in about 50% of prostate cancers [54]. Currently, however, it is not known whether a similar translocation takes place in a subset of lung cancers; indeed, the TMPRSS2–ERG fusion gene seems to be specific to prostate cancer, which may be due to the strong induction of TMPRSS2 by androgen [55]. A series of more than 60,000 cancer cases was used to determine the frequency of the TMPRSS2–ERG fusion, assayed by comprehensive genomic profiling (CGP). The fusion gene was detected exclusively in tumor samples from male patients, 30% of which were classified as prostatic cancers. Furthermore, the TMPRSS2–ERG gene was also identified in four cases of lung cancer, four cases of bladder cancer, and two cases of pancreatic cancer [56]. Unfortunately, this seems to be a lone study demonstrating the presence of the TMPRSS2–ERG fusion gene in non-prostatic cancers. Further analysis might be done to demonstrate if the presence of this AR-related gene can be used in other AR-positive cancers. The studies so far presented highlight the role of the androgen receptor as a negative prognostic factor in the onset of lung cancer; moreover, the AR could be used as a molecular target in cancer subtypes that overexpress it, such as NSCLC. It is important, therefore, to further investigate the involvement of this receptor in lung cancer.

Although lung cancer has long been a disease characterized by late-stage diagnosis and no progress in treatment options, in the last decade, lung cancer screening in high-risk populations has yielded encouraging results and substantial progress has been made with personalized therapies chosen on the basis of cancer subtype and stage.

To bolster this new and effective approach to lung cancer therapy, it is mandatory to discover new molecular targets, and the AR could represent a good candidate. Gockel et al. have exploited the ubiquitin–proteasome system with proteolysis-targeted chimeras (PROTACs) to degrade ARs in lung cancer cells [57]. The PROTACs, based on the use of the anti-androgen enzalutamide, were able to robustly induce AR degradation in lung cancer cells [57]. Considering that the reduction of AR levels represents, for many scientists, a way to fight lung cancer [50,58], this technique could be an innovative and promising therapeutic strategy. A similar approach is currently used in prostate cancer patients with beneficial results [59–61].

These findings have stimulated research in investigating new drugs targeting the AR in monotherapies or therapies combined with the currently available antiandrogens in the management of lung cancer patients.

## 4. AR in Kidney Cancer

Kidney cancer, also named renal cancer, includes different histologically defined subtypes. Renal cell carcinoma (RCC) accounts for 90–95% of all cases, and transitional cell cancer (TCC) and Wilms' tumor represent the remaining 5–10% of cases [62]. The lack of premonitory symptoms and diverse clinical manifestations make RCC a neoplasia difficult to diagnose and treat and, with 431,888 cases registered in 2020, this tumor is the 14th most common cancer worldwide; the 9th most common cancer in men and the 14th in women (https://gco.iarc.fr/today/data/factsheets/cancers/29-Kidney-fact-sheet.pdf, accessed on 15 February 2023). Worldwide incidence and mortality rates are steadily

increasing at a percentage of approximately 2–3% per decade [63], and in 2023, in the United States, there will be registered an increase of 5% and 3% of new cases in men and women, respectively [45]. The RCC etiology is still largely unknown, although determinants such as obesity, diabetes, hypertension, cigarette smoking, and diet have been reported as potential risk factors [64,65].

RCC is not generally considered a sex-hormone-dependent cancer, but recent studies have indicated that a gender difference exists, with a male to female incidence ratio of 1.6 to 1.0 [65]. This difference might suggest that sex hormones and/or their receptors are involved in the development of RCC. However, there are few studies that relate sex steroid receptors such as the AR to the initiation and progression of RCC, and the published studies show conflicting results.

The AR influences proliferation, migration, and invasion in in vitro and in vivo models of RCC by altering the hypoxia-inducible factor $2\alpha$ (HIF-$2\alpha$)/vascular endothelial growth factor (VEGF) pathway at the mRNA and protein expression level. Treatment of human AR-positive RCC cell lines with DHT increased their proliferation, migration, and invasion, whereas the use of ASC-J9, an AR degrader, suppressed proliferation in vitro and reduced xenograft growth in vivo [66].

Other studies exploring the role of the AR in the most frequent subtype of RCC, clear cell RCC (ccRCC), correlate this receptor to a poor prognosis. Deleted in Breast Cancer 1 (DBC1) is related to poor prognosis in many cancers, despite its first identification as a tumor suppressor. ARs were expressed in 163 of 200 human-derived RCC specimens, without differences between male and female patients. AR expression positively correlated with p53, histone deacetylase sirtuin 1, SIRT1, and DBC1 expression, and all those proteins resulted in poor prognosis and survival markers. DBC1 positively modulated ARs, activating cell signaling pathways and driving tumor progression. In sum, while SIRT1 and p53 can be related to the pathogenesis of renal cancer, the DPC1-AR pathway could be involved in the carcinogenesis and progression of this disease. [67].

In contrast with these studies, Protein Atlas considers AR, which is detected in renal epithelial cells, a favorable prognostic marker in RCC. Statistical analyses show that AR expression in cancer is positively related to a better cancer survival. Patients in this study, however, were not analyzed for sex or age (The Human Protein Atlas. Available online: https://www.proteinatlas.org, accessed on 17 February 2023)

Nonetheless, analyses conducted with the GENT2 database showed no differences in AR expression between normal and tumor tissues (GENT2. Available online: http://gent2.appex.kr/gent2/, accessed on 17 February 2023).

Recently, Bialek and colleagues demonstrated differences in the AR and its splicing variants' (AR-V1, -V3, -V4 and -V7) expression between the two main types of RCC—ccRCC (clear cell RCC) and pcRCC (papillary cell RCC)—and between the pathological pT stages of ccRCC tumors [68]. In addition, a potential modulator of AR, relaxin (RLN2), was evaluated. The AR and its splicing variants are more highly expressed in pcRCCs than in ccRCCs, both in tumor and in normal paired tissues. Furthermore, this study highlighted an inverse correlation between the expression of the AR and its variants and tumor stage in ccRCC. In addition, RLN2 expression and tumor growth are negatively related, while RLN2 and AR expression show a significantly positive correlation in male patients. These data suggest the possibility of indirect or direct dependence between RLN2 and ARs in renal carcinoma, especially in men with ccRCC, supporting a favorable role for the AR in RCC [68].

Consistent with these findings, other studies have correlated AR expression to male patients, early-stage and low-grade tumors (Fuhrman), moderate differentiation, and good prognosis [69,70].

Nevertheless, the role of the AR in RCC is still controversial, likely because AR-related signaling is a complex and multi-stage process that may influence other responses such as inflammation, EMT, migration, or cell proliferation, which are crucial for tumor development and metastasis.

Therefore, molecular analysis of RCC and a better understanding of the disease are crucial to increase the potential for personalized treatment in these patients.

Although the development of new therapies targeting vascular endothelial growth factor (VEGF) and tyrosine kinases open new doors for patients with advanced RCC, their effect is still limited for patients with selective disease types [71,72]. Therefore, surgery remains the only effective treatment for RCC, since metastatic disease is usually resistant to radiotherapy and chemotherapy, and immunotherapy shows limited response rates ranging from 15% to 20% [73]. The search for new and improved therapies for metastatic RCC is still needed. Therefore, although controversial, the role of the AR should be studied and research expanded to understand the signaling pathways that can be activated and identify this protein as a diagnostic and therapeutic marker in renal carcinomas.

## 5. AR in Bladder Cancer

Bladder cancer (BCa) is the most common neoplasm of the urinary tract, and one of the most common cancers in the world [74].

The main risk factors driving BCa are age, cigarette smoking, alcohol, obesity, and excessive red meat use [75]. Recently, it has been shown that chemicals such as aromatic amines and aniline dyes may also contribute to the occurrence of BCa [76].

The incidence of this cancer is 3–4 fold higher in men than in women [45] and, although the etiology of this difference remains unknown, hormonal differences have been considered as a potential explanation for this gender disparity [77]. Studies in animal models have shown that the incidence of spontaneous and chemically induced bladder tumors is significantly higher in male rats than in females. In addition, treatment with androgen deprivation therapy has been shown to reduce the development of chemically induced BCas [78,79]. These findings encourage further investigation of the role of androgens and ARs in this cancer. The physiological functions of androgens and ARs in the bladder are unclear, and only a few studies have shown that AR signaling contributes to the regulation of urine storage and cholinergic as well noncholinergic nerve functions in the urinary tract [80,81]. However, the role of the AR in bladder cancer development remains unexplored, with scant information in the literature.

Few studies, with conflicting results, have analyzed the role of the AR in BCa, likely because of the different investigation methods. Thus, the AR's functions and molecular mechanisms in this cancer still remain unclear.

Laor and colleagues reported that the expression of the AR is higher in bladder cancer than in healthy bladder mucosa [82]. By contrast, high-grade tumors exhibit less ARs than low-grade tumors [82]. In support of this evidence, more recent studies by Boorjian and co-workers has shown that decreased AR expression is associated with increased pathologic stage and that the loss of AR expression is associated with invasive bladder cancer [83].

These data make unclear the role of the AR in bladder cancer growth and seeding but might explain the incidence disparity. AR expression could be essential during the first stages of bladder cancerogenesis, and only the AR-positive cells could be exposed to transformation.

As in many other cancers, DHT-activated ARs trigger a plethora of signaling pathways by both genomic and non-genomic actions in BCa [84,85].

The high frequency of AR expression in bladder cancer cells might explain the prevalence of this tumor in men and suggest the use of AR as potential marker for this cancer. AR expression is often correlated to the expression of p53 and c-erb-2, which are both related to the invasiveness, stage, and histology of this neoplasia. These observations could make AR a possible bladder tumor marker [86,87].

Activation of the AR-dependent pathway is due to several factors, including the inflammatory cytokine IL-8. The latter is produced and released by B lymphocytes in the tumor microenvironment of BCa. Experiments performed on three BCa-derived cell lines co-cultured with B lymphocytes showed increased invasion and metastasis of tumor cells, and this phenomenon might be due to the ability of IL8 to influence AR signaling. In particular,

the presence of IL8-activated ARs in BCa cells has been linked to the ability of ARs to control the transcription of different metastatic genes, including the metalloproteinases 1 and 13 [88], which likely increase cell migration and invasion.

Many studies demonstrate a strong correlation between androgen deprivation therapy (ADT) and BCa incidence. ADT for prostate cancer was associated with a decreased risk of bladder cancer and seems to be a promising therapy for lowering BCa recurrence [89–92]. Other studies failed to identify any impact of ADT on the risk of developing BCa [93,94], but this could be due to several limitations in those observational analyses [94]. For this reason, future methodologically rigorous studies addressing the limitations underlined by Santella and colleagues are needed to evaluate the important potential association between ADT and bladder cancer [94].

The commonly used therapy for BCa is chemotherapy; however, it has been seen that a significant number of patients with urothelial cancer do not respond to systemic cisplatin-based chemotherapy. The mechanisms underlying chemoresistance remain poorly understood, although in vitro studies have recently suggested the relationship between AR activity in urothelial cancer cells and chemoresistance to cisplatin [95] or gemcitabine [96]. In both cases, chemoresistance is associated with a higher AR expression in BCa cells, leading to the hypothesis that the AR drives this process.

Thus, androgen deprivation therapy, which is widely used for prostate cancer treatment, could be a possible alternative or, more likely, an adjuvant to chemotherapy in those cancer cells that express the AR [97].

The presence of ARs in this kind of tumor appears contradictory. Some studies point to a critical role of androgen-mediated AR signaling pathways in urothelial carcinoma pathogenesis and metastasis progression, supporting the idea that it represents an endocrine-related neoplasm. Other studies, however, have argued that the receptor is unable to drive metastasis given its low expression in high-stage and metastatic BCa cells.

In sum, it seems important to continuously investigate the role of this receptor and its dependent signaling pathways to fully understand its function and explore the opportunity to use the receptor as a prognostic factor or molecular target for new therapies.

## 6. AR in Hepatocarcinoma

Hepatocarcinoma (HCC) is the fourth leading cause of cancer deaths worldwide and the sixth most common cancer. HCC is a sexually dimorphic cancer, with a 2–7 fold higher incidence rate in men than women [45,98]. For this reason, it appears interesting to investigate the role of the AR in this cancer. Different studies with conflicting or undefined results have pointed to ways to clarify the relationship between the AR and HCC prognoses.

A study from Acosta-Lopez and colleagues demonstrated that, while the AR expression is a favorable marker for HCC prognosis, its activity is associated with poor prognosis in patients with HCC [99].

Wu et al. suggested that the AR plays a key role in hepatitis-B-virus-induced HCC by promoting HBV transcription and therefore hepatocarcinogenesis [100]. This activation cannot explain the gender difference of both viral and non-viral HCC but suggests that targeting of AR might represent a new therapeutical strategy to prevent HBV-induced hepatocarcinogenesis.

Zhang et al. focused their report on clarifying the role of mTOR in the induction of AR expression in HCC [101,102]. It was already known that mTOR is overactive in HCC and plays a key role in the promotion of the metabolic activity, proliferation, and survival of tumors [103]. Studies by Ren and Zhang have shown a relationship between the AKT/mTOR pathway and the AR in HCC [101,102]. In detail, the mTORC1 factor inhibits AR degradation by the phosphorylation of a serine residue in position 96 and increases its nuclear translocation. Once in the nucleus, the AR promotes the expression of FKBPS, which interacts with PHLPP1, a phosphatase of AKT, to inhibit AKT phosphorylation and extend the mTORC1 activity, thereby setting up a feedback mechanism [101,102]. These

findings allow us to consider co-targeting the AR and mTOR as a potential therapeutic strategy for HCC treatment.

In another report, Cheng and Colleagues presented a different relationship between the AKT/mTOR pathway and the AR in HCC mediated by non-alcoholic hepatic steatosis, and they discovered how AKT mediates the activation of AR by diacylglycerols (DAGs) [104].

On the other hand, additional reports have shown that the AR promotes the progression of HCC through the over-regulation of cell-cycle-related kinase (CCRK), resulting in the upregulation of β-catenin and, consequently, of the AR, thus making a feedback mechanism. Over-regulation of CCRK, β-catenin, and the AR is associated with a poor prognosis in HCC patients [105]. Another feedback mechanism between the AR and CCRK operating during HCC progression involves the engagement of the STAT3 protein. It seems that CCRK not only promotes the interaction between STAT3 and ARs, but also the localization of this complex on the ARE, located in the CCRK promoter itself, which indirectly activates the mTOR pathway and elicits HCC progression [106].

At least, AR expression was positively correlated with the expression of TLR4, a factor that promotes the proliferation of HCC and induces the production of pro-inflammatory molecules related to HCC malignancy [107].

The use of AR antagonists in HCC as a possible therapeutic strategy has been also tested, with disappointing results [108,109]. Jiang and colleagues reported about the potential impact of ARs on the cancer microenvironment and immune surveillance in HCC and demonstrated how the AR directly interacts with programmed death ligand-1 (PDL-1) by reducing its expression, altering the tumor microenvironment, and enhancing the function and proliferation of activated CD8+ T cells. This scenario indicates that the different expression of the AR in HCC cells may provoke shifts in the immune response, thus opening the way for the development of new immunotherapeutic strategies for HCC [110].

## 7. AR in Pancreatic Ductal Adenocarcinoma

Pancreatic cancer (PaC) is a rare lethal disease including two different subtypes: exocrine and the neuroendocrine pancreatic cancer. Exocrine PaC represents 95% of all PaCs, and the most represented subtype of this group is pancreatic ductal adenocarcinoma (PDAC), a devastating disease with a median global survival time of 5 months and a less than 5% five-year survival rate. PDAC has a male–female incidence ratio ranging from 1.25 to 1.75:1 [111] and approaches a 1:1 ratio with aging [112]. Similar ratios are projected to occur in United States in 2023 [45].

The AR expression in PDAC was initially questioned, but subsequent studies confirmed that pancreatic cancer cell lines expressed variable levels of ARs [113,114]. Further in vitro experiments have shown that PDAC tumor cells responded in a different way to treatment with an AR agonist, showing a modest increase in cell proliferation [113,115]. In light of this, the effect of flutamide, an AR antagonist used in prostate cancer treatment, was tested on PDAC cell proliferation and it was observed that its effect on cell proliferation reduction did not correlate with AR expression [113]. Furthermore, treatment with flutamide did not change the cellular response to the generic anticancer drug gemcitabine either in vitro and in vivo [113]. Successive reports evaluated the effect induced by EZN, an AR inhibitor with higher affinity and antagonistic activity, used in combination with classic anticancer drugs as therapy in patients with metastatic pancreatic cancer. The combined treatment of PDAC patients with gemcitabine, nab-paclitaxel, and enzalutamide was well tolerated in a phase I trial, and it robustly reduced the level of the PaC marker, CA19-9. Furthermore, the combo treatment prolonged the overall survival as well as the disease-free survival [116]. ENZ, therefore, represents a promising adjuvant drug in patients with metastatic pancreatic cancer.

Additional reports showed that AR activity is modulated by IL-6, an inflammatory cytokine overexpressed in pancreatic carcinoma [117].

IL-6 is, indeed, were involved in the trans-activation of AR through the STAT3 and MAPK pathways [118], and previous findings from Ueda and colleagues demonstrated a direct interaction between the N-terminal domain of the AR and STAT3 beyond IL-6 treatment in prostate cancer [119]. A similar mechanism might explain the AR activation induced by IL-6 in pancreatic cancer cell lines. Furthermore, in the same report, Okitsu et al. revealed that IL6 promoted pancreatic cancer cell migration in the presence of ARs [118], making the receptor a putative therapeutic target in pancreatic cancer treatments.

## 8. AR in Gastric Cancer

Gastric cancer (GC) is the second most frequent cause of cancer deaths worldwide and the fourth most diffuse cancer. About 90% of these tumors are adenocarcinomas derived from epithelial cells of the gastric mucosa, difficult to diagnose at an early stage because of the paucity of symptoms. The disease is detected in an advanced stage and tends to have a poor prognosis. Unfortunately, there are no specific therapies for GC [120,121], and surgery still remains the only chance to cure this tumor [122].

For these reasons, there is a need to investigate new prognostic markers as well as more successful therapies for severe GC. Different reports have indicated that the incidence of GC is significantly higher in men than in women, with a ratio of 2:1 [123], and this will be similar in 2023 [45]. In the 1990s, Wu and colleagues demonstrated for the first time the significant presence of ARs in GC [124], so later reports have since focused on studying the prognostic and therapeutic role of this receptor in GC.

AR overexpression promoted cell migration and invasion in both in vivo and in vitro models of GC. To understand the mechanisms behind these effects, several effectors were analyzed, including metalloproteinase 9 (MMP9), which is transcriptionally upregulated by AR as a consequence of a direct interaction with the gene promoter [125].

Tang and colleagues confirmed that AR activity promotes migration, invasiveness, and EMT of cultured GC-derived cells through the upregulation of proteins directly responsible for these processes, such as β-catenin, snail, slug, and alpha smooth muscle actin (α-SMA, [126]). Fard and his research team focused, instead, on the analysis of the effect of AR inhibitors such ENZ on cell proliferation and the EMT in GC [123]. They found that the AR is expressed at a higher level in GC with an advanced TNM stage and its expression is positively correlated to β-catenin and negatively to E-cadherin, while they observed a moderate positive correlation between AR and other EMT markers such as snail, twist1, and STAT3. They found that ENZ affected the G2/M transition and caused apoptosis by downregulating cyclin-B1 and Cdk1 and upregulating p21 and caspase-3 in GC cell lines. Furthermore, ENZ inhibited cell migration and invasion by inhibiting the closure in wound healing assays and reducing the MMP9 and MMP2 activities. EZN was also able to reduce EMT, reducing the β-catenin, snail, twist1, and STAT3 levels and increasing the E-cadherin level. In addition, the authors observed that the addition of ENZ to 5-fluorouracil (5-FU), an effective chemotherapeutic drug used in GC therapy, increased the effectiveness and the cytotoxicity of 5-FU [123].

Additionally, in a different report, the same investigators discovered that about 85% of GC patients who overexpress AR also overexpress Aurora kinase A (AURKA), an important mitotic kinase, which plays a key role in the regulation of the cell cycle and several oncogenic pathways. This report suggests that the co-existence of both the proteins AR and AURKA could be used as a prognostic factor in GCs [127].

In advanced stages of GC, ARs might promote chemoresistance to cisplatin by inducing the activation of laminin subunit alpha 4 (LAMA4) through direct interaction with its promoter [128]. LAMA4 is generally upregulated in cancer, and its high expression level is related to gemcitabine, cisplatin, trastuzumab, Adriamycin, and vincristine resistance in GC. The ability of the AR to promote chemoresistance by increasing LAMA4 expression suggests that therapeutical strategies targeting the AR and LAMA4 itself could improve the drug response of chemotherapy-resistant GCs.

## 9. Discussion and Concluding Remarks

The data reported in this review and illustrated in Figure 3 have summarized the role of the AR in the development and progression of tumors that are not classically considered hormone-dependent, but show a gender-related incidence, which is much higher in males than in females.

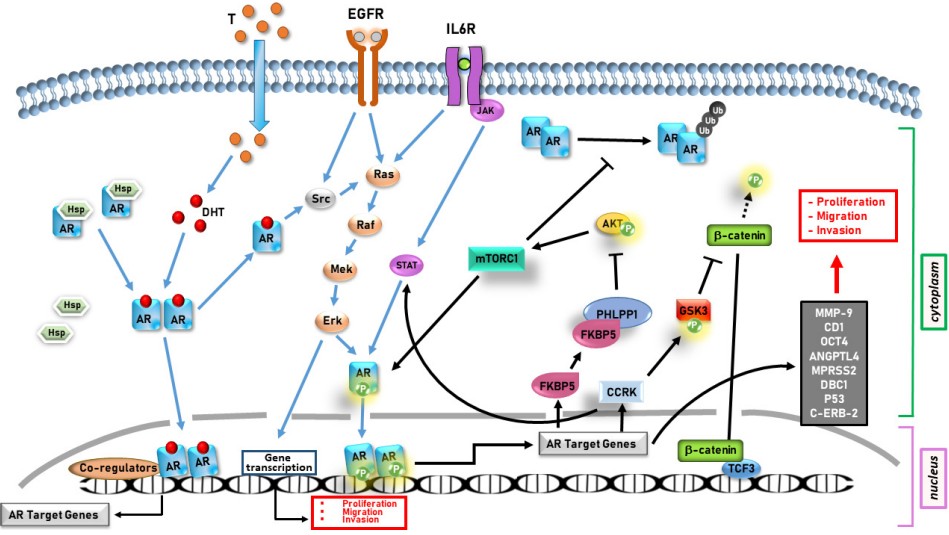

**Figure 3.** An overview of AR signaling in gender-related cancers.

In liver, lung, pancreatic, and gastric cancers, the AR supports tumor progression, whereas in bladder and kidney cancers, its role is still controversial.

In particular, in hepatocarcinoma, characterized by a marked sexual dimorphism [98], the AR induces cell proliferation through the upregulation of cyclin kinase CCRK [105,106] and tumor progression through activation of the AKT/mTOR signaling pathway [101,102].

In PDAC, although the male-to-female ratio is 1.25/1.75:1, and thus lower [111], the AR still plays an important role in promoting cell migration and invasion by activating the AR/IL-6/STAT3/MAPK signaling pathway [118].

In lung cancer, which has a male–female incidence ratio of 1.9:1 (http://globocan.iarc. fr/, accessed on 11 March 2023 ), the AR promotes cell proliferation through the activation of both the transcription of cyclin D1 and the signaling controlled by the CD1-mTOR pathway [50,51]. Furthermore, the AR promotes metastasis formation by upregulating OCT4, ANGPTL4, and MPRSS2 [48,52].

In gastric cancer, whose incidence is twice higher in males than in females [123], the AR upregulates the gene that codifies metalloprotease 9 (MMP 9), thus promoting cell invasion [125]. Furthermore, by using the antagonist ENZ, it has been demonstrated how the AR correlates with cell proliferation and survival [123].

More difficult is to draw conclusions about the role of the AR in kidney cancer, whose male–female incidence ratio is 1.6:1 [65]. It has been found that the AR facilitates cell proliferation, migration, and invasion through the factors HIF-2α, VEGF, and DBC1, on one hand [66,67]. On the other hand, relaxin-related AR expression could be an important positive prognostic factor [68].

Even in bladder cancer, which is characterized by a 3/4:1 male-to-female ratio [77], the AR has a controversial role. This receptor is highly expressed in low-stage tumor tissues, but its levels are reduced with advancing disease [82,83]. However, the AR has been found to play an important role in the progression of this tumor, as it causes an increase in the expression of the oncogenes p53 and c-erb-2 [86] and it is implicated in cell invasion and metastasis formation through the activation of IL-8/AR signaling [88]. In sum, the collected data highlight the role of the AR in gender-related cancer (Figure 4).

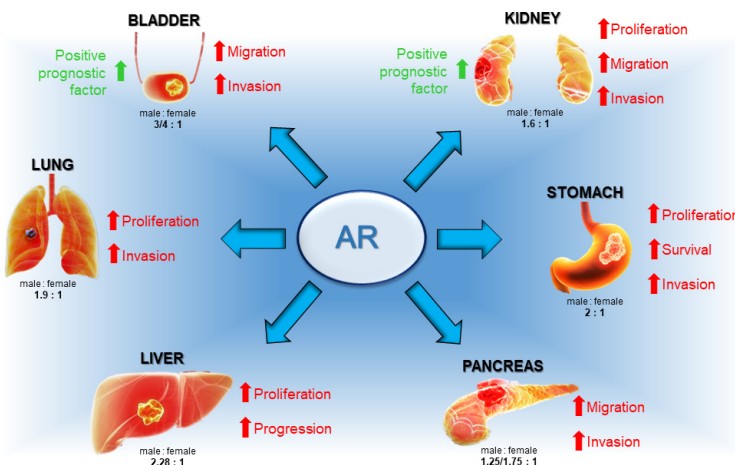

**Figure 4.** Main actions of AR in gender-related cancers. Whereas in lung, liver, pancreas, and stomach cancers AR contributes to tumor development and progression, in bladder and kidney cancers, the receptor works in two opposite ways by both promoting and inhibiting cancer.

In the majority of the discussed cancers, the AR works as a promoter and supporter in tumorigenesis and tumor progression, although further studies are necessary to understand if it can be used as a marker in diagnosis and treatment. In other cases, mostly in bladder and gastric cancers, the role of the AR is still controversial, and additional data are indispensable to clarify its role in these cancers. Anyway, it would be interesting to analyze both the expression and potential role of AR isoforms and splicing variants in these gender-related cancers, especially when the functions of the AR are not completely clear. AR variants might work in an antagonistic way, thus explaining the conflicting data so far obtained on gender-related cancer [18,20,21] but, to date, there are scant options for detecting and discerning between the AR and its modified versions. Furthermore, the literature only describes the genomic action of the AR and never investigates any non-genomic mechanism involved in the oncogenesis or progression of this group of cancers. The study of these mechanisms together with the development of techniques capable of accurately observing the expression of the AR and its variants might strongly increase knowledge in the unexplored field of the androgens' influence in gender-related cancers. Depending on its specific role in gender-related cancers, new therapies can be developed by using the AR as target. When the AR works as a tumor inhibitor, the use of selective androgen receptor agonists (SARMs), activating the receptor with a high specificity and affinity, could represent an effective therapeutic strategy, mostly related to the existence of reduced side effects. Furthermore, these SARMs can be used alone or in combination with other therapies, such as immunotherapy, to increase their efficiency, as was successfully demonstrated in AR-positive metastatic TNBC patients (NCT02971761; [129]) or in AR/ER-positive BC (NCT03088527; [129]). On the other side, the use of AR antagonists could be a therapy in all the gender-related cancers in which the AR promotes disease progression. To this purpose, many data confirm the efficiency of Enzalutamide alone or in combination with chemotherapeutic or immunotherapeutic agents in in vitro [96,130] and in vivo experiments [131] and also in clinical trials on gender-related cancers. In addition to antiandrogens, therapies with selective AR degraders (SARD) could represent a revolutionary strategy to cure all the gender-related cancers controlled by AR isoforms resistant to the classical antiandrogens [129].

Statins, too, could represent an acceptable therapy for gender-related cancers. Statins reduce cholesterol levels, and this lipidic molecule is the sex steroid precursor. The effects of this therapy should be evaluated for every single tumor or patient, considering that when different steroid receptors are simultaneously expressed, they could work in opposition to each other, as in ER-positive breast cancer expressing ARs [12]. Anyway, some data in the literature demonstrate that statins, alone or in combination with other chemotherapeutics,

can control the growth of different types of cancers such as breast, ovarian, pancreatic, and lung cancers [132–140].

Understanding how androgens and their receptors influence gender-related cancer is important not only for developing new therapeutic strategies, but also for identifying at-risk subjects and organizing prevention campaigns for lowering cancer incidence. For example, analyzing the incidence of these cancers in transgender individuals or in subjects with altered androgen concentrations, such as women with polycystic ovary syndrome (PCOS), might produce useful results. Currently, several studies have characterized the incidence of cancers in transgender patients, but the main attention was paid to breast and prostate cancers in transmen and transwomen, respectively [141–144]. An analysis conducted on bladder cancer transgender patients observed a higher incidence in transgender individuals compared with ciswomen, but not with cismen [141]. On the other hand, another study demonstrated that ciswomen had a 5-year survival rate that was lower than cismen [145]. Furthermore, in animal studies, the observed sex difference in bladder cancer carcinogenesis disappeared after male mice castration and testosterone administration to female mice [146]. These data confirm that sex hormones play a role in gender-related cancers and show that testosterone promotes its carcinogenesis, whereas estrogens appear to inhibit carcinogenesis, but promote tumor progression [147,148].

Recently, the relationship between PCOS and other cancers, such as PaC or RCC, was analyzed and confirmed by many studies [149–152]. These results suggest that diagnosis of PCOS may warrant increased education and clinical vigilance for PaC, but additional studies are required.

All the results discussed in this review highlight the need to understand the role of androgens and their receptor in gender-related cancers in order to reduce their incidence and mortality by drawing up both preventive and therapeutic plans.

**Author Contributions:** Conceptualization, P.G., E.S. and V.T.; writing—original draft preparation, P.G., E.S. and V.T.; writing—review and editing, P.G., A.M. and G.C.; visualization, V.T., E.S. and F.L.; supervision, G.C. and A.M.; funding acquisition, G.C., A.M. and M.D.D. All authors have read and agreed to the published version of the manuscript.

**Funding:** This research was funded by the Italian Ministry of University and Scientific Research (P.R.I.N. 2017EKMFTN_002 to G.C.), Regione Sicilia (Progetto di Ricerca Finalizzata RF—2019—12368937 to A.M.) and Vanvitelli Young Researcher (PATG.Rice.Base.Giovani Ricercatori2022.IDEA to M.D.D).

**Conflicts of Interest:** The authors declare no conflict of interest.

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
