# Peer review of "Role of the Androgen Receptor in Gender-Related Cancers"

_endocrines, doi:10.3390/endocrines4020031_

Round 1

Reviewer 1 Report

This paper presents an overview of the role of the androgen receptor (AR) in hormone-dependent cancers, including prostate and breast cancer, as well as non-classical hormone-dependent cancers. The authors examine the roles of AR in regulating tumor development and progression and suggest that gaining a better understanding of AR signaling mechanisms could lead to the development of novel diagnostic and therapeutic strategies for gender-related cancers.

1, It may be beneficial for the authors to provide a clear definition or classification of non-classical hormone-dependent cancers to avoid any potential confusion for readers. While the title of the paper suggests a focus on gender-related cancers, specifying the classification of non-classical hormone-dependent cancers would improve the reader's understanding of the content.

2, The authors could expand upon the possible diagnostic and therapeutic approaches that may arise from a deeper understanding of AR signaling in gender-related cancers and highlight any associated challenges.

3,  It should also be noted that AR has been reported to play a role in certain gender-related cancers in transgender individuals undergoing gender-affirming hormone therapy with exogenous testosterone. Further discussion on this topic may enhance the paper's comprehensiveness.

Author Response

We would like to thank the reviewer for giving time and reviewing our article and we strongly appreciate your feedback that helped us improve our manuscript. Below you will find the answer to each question as well as the information about the changes inserted in the manuscript. All changes are highlighted.

We tried our best to respond to each concern raised.

REVIEWER #1. This paper presents an overview of the role of the androgen receptor (AR) in hormone-dependent cancers, including prostate and breast cancer, as well as non-classical hormone-dependent cancers. The authors examine the roles of AR in regulating tumor development and progression and suggest that gaining a better understanding of AR signaling mechanisms could lead to the development of novel diagnostic and therapeutic strategies for gender-related cancers.

  1. It may be beneficial for the authors to provide a clear definition or classification of non-classical hormone-dependent cancers to avoid any potential confusion for readers. While the title of the paper suggests a focus on gender-related cancers, specifying the classification of non-classical hormone-dependent cancers would improve the reader's understanding of the content.

We thank the reviewer for this observation. In general, all the cancers of which growth and invasiveness are notoriously and directly controlled by sex-steroid hormones and their receptors are known as hormone-related cancers such as breast cancer in women and prostate cancer in men. Other examples are testicular cancer in men, and uterine and ovarian cancers in women. Except for breast cancer, occurring in both sexes with a clear predominance in women, all these cancers are also sex-specific.

With “non-classical hormone-dependent cancers” we indicate all those cancers occurring in both sexes, for which it is unknown any dependence from sex steroid hormones and receptors but exists an incidence imbalance between men and women. This gap could be explained by the unequal concentration of circulating hormones between men and women. This last group is also indicated as “gender-related cancers” in our manuscript.

We explained this in the text, Abstract lines 10-13; Introduction, lines 70-83, yellow text marker.

  1. The authors could expand upon the possible diagnostic and therapeutic approaches that may arise from a deeper understanding of AR signaling in gender-related cancers and highlight any associated challenges.

Diagnostic and therapeutic approaches in cancers where AR appears to be predominantly a negative factor, such as lung, stomach, liver, and pancreas, could be given by investigating AR expression in tumor tissues and using combination therapies with classic chemotherapeutics. The choice of using selective degraders or inhibitors of AR or other proteins involved in AR signaling involved in tumor progression could be a novel therapeutic approach in these tumors. On the other hand, as the role of AR is still unclear for kidney and bladder cancer, it is preferable to continue to investigate to best understand how diagnostic and therapeutic action can be taken.

We explained better this in Discussion and Concluding Remarks, lines 608-623 (yellow text marker)

  1. It should also be noted that AR has been reported to play a role in certain gender-related cancers in transgender individuals undergoing gender-affirming hormone therapy with exogenous testosterone. Further discussion on this topic may enhance the paper's comprehensiveness.

We deeply appreciate the reviewer’s comment. This topic is very important to us but, unfortunately, in literature, the number of research about it is limited and not sufficient to write an exhaustive and complete paragraph. Conversely to the study about hormone-dependent cancer (breast, prostate, cervix, and others), the study of gender-related cancers in transgender individuals is only collaterally mentioned in a few publications. Anyway, we added some data and consideration to the Discussion and Concluding Remarks, lines 634-647, yellow text marker.

Reviewer 2 Report

In the current review, authors have done a great job describing the roles of AR in the cancers besides prostate and breast cancers, including lung, kidney, bladder, hepatocarcinoma, pancreatic ductal adenocarcinoma, and gastric cancer. The review study compared the role of AR in the aspect of genders. It provided a valuable overview of AR function in gender-related cancers.

A few questions, 

1.     Line 77, AR gene (isoform 1) is a ~11 kb and encodes a 110 kDa protein. 

2.     When authors describing AR gene isoforms and variants, I recommend combining the current description of isoforms and variants and follow the information from NCBI, that there are 5 isoforms.

3.     There are many point mutations, such as, AR-Q865H, they don’t have to be introduced as one of the general AR gene variants.

Author Response

We would like to thank the reviewers for giving time and reviewing our article and we strongly appreciate your feedback that helped us improve our manuscript. Below you will find the answer to each question as well as the information about the changes inserted in the manuscript. All changes are highlighted.

We tried our best to respond to each concern raised.

REVIEWER #2. In the current review, authors have done a great job describing the roles of AR in cancers besides prostate and breast cancers, including lung, kidney, bladder, hepatocarcinoma, pancreatic ductal adenocarcinoma, and gastric cancer. The review study compared the role of AR in the aspect of gender. It provided a valuable overview of AR function in gender-related cancers.

A few questions,

  1. Line 77, AR gene (isoform 1) is a ~11 kb and encodes a 110 kDa protein.

We thank the reviewer for this observation and apologize for the mistake. We substituted the wrong unit of measurement with the right one (lines 92-93, green text marker).

  1. When authors describe AR gene isoforms and variants, I recommend combining the current description of isoforms and variants and following the information from NCBI, that there are 5 isoforms.

Five isoforms are shown on the NCBI site, of which isoform 1 is named in the manuscript isoform B and isoform 2 is isoform A. The other isoforms on the NCBI.gov site, isoforms 3, 4, and 5, are splicing variants; in particular, isoform 3 is mentioned in the manuscript as splicing variant AR-V7, isoform 4 is named AR-V1, and isoform 5 corresponds to splicing variant AR8. We added this classification to the text, lines 149-151, green text marker.

  1. There are many point mutations, such as AR-Q865H, they don’t have to be introduced as one of the general AR gene variants.

We deleted this genetic variant in Fig.2.

Reviewer 3 Report

Sabbatino et al review literature on androgen receptor action in gender-related cancers. This article highlights the role of the AR in prostate, breast, lung, and gastrointestinal cancers. The authors discuss how AR signaling contributes to cancer onset and progression. They also suggest that understanding the role of AR in these tumors could help identify new biomarkers for early diagnosis and develop better therapeutic approaches by directly targeting AR or its downstream signaling. Overall, this paper highlights the importance of studying AR expression in non-classical hormone-dependent cancers to improve cancer treatment options. This review would benefit from more synthetization of the literature to flow better; it is choppy and disjointed in certain sections (as discussed below). Additionally, this manuscript would benefit from further comment/discussion of the potential implications of AR targeting in various disease stages. Lastly, the article has numerous grammatical and spelling errors that need attention. Please consider addressing the following concerns.

Major Comments:

1)    Please comment of what drives expression of different AR isoforms – A vs B. Is their expression gender-related? Disease specific? Organ, tissue, or cell type specific or highly favored? How well does each isoform respond to androgens and anti-androgens?

2)    On line 118, the authors mention how AR mutations play a role in oncogenic signaling. Please comment on key mutations observed to be oncogenic drivers.

3)    What other types of cancers have the TMPRSS2-ERG fusions? Are they AR dependent in those as well?

4)    What is the gender ratio of incidence and survival outcome in lung cancer? Please add to the lung cancer section.

5)    Consider rearrangement of the Renal Cancer section by having an intro paragraph on stats and different types of renal cancer. Then go into the studies depicting positive roles of AR and then negative roles of AR. Mention the gender differences or the gaps that remain in this disease type to further discern the role of AR in renal cancer.

6)    What is the prevalence of patients who have PCa treated with anti-androgens and/or AR antagonists that develop bladder cancer? Is this less than patients that never receive any prior AR targeted therapies? What does this suggest about the role of AR in other cancer types beyond prostate and gender-bias ratios observed?

7)    How do AR targeted therapies directed at dampening the cholesterol biosynthesis pathway impact other hormone nuclear receptors in breast, ovarian, lung cancer, etc? Would this be a beneficial type of treatment for gender-related cancers beyond AR targeting?

8)    What is the take-home message? Does gender impact and/or dictate the function of AR in cancers? Or is AR’s role independent from gender and its correlation with incidence and outcome is associated with other contributing factors?

9)    Please fix grammatical and spelling errors present throughout manuscript. Also make sure one sentence does not constitute a paragraph. Please fix throughout manuscript to combine single statements into complete paragraphs.

Minor Comments:

1)    Please define AR when utilizing it for the first time in the text on line 56.

2)    Prostate Cancer abbreviation is “PCa”. PC is often utilized to abbreviate pancreatic cancer.

3)    Define NTD on line 80.

4)    Please keep consistent in utilizing acronyms once they have been defined throughout the text.

5)    On line 142, the authors only reference one ovarian cancer and AR paper, which is a review article. Please make sure to credit the additional studies done on AR in this specific cancer type.

6)    Make sure to add proper references in text. For instance, on line 165 refer to “Mikkonen and colleague”" but reference is not added. Additionally, on lines 173-174 “Recchia et al” reference is not added. Please thoroughly check the entire manuscript and fix the references. This happens in several places.

7)    Please update cancer statistics throughout manuscript with 2023 stats.

8)    Make sure to italicize gene names and “in vitro” and “in vivo” terminology throughout the text.

9)    In Fig 3, please also show that AR interacting with co-regulators leads to alterations in target gene expression. Additionally, SRC kinase also regulates AR and induces a kinase signaling cascade. Consider adding to the schematic. Lastly, add labels of “cytoplasm” and “nucleus” to schematic.

Author Response

We would like to thank the reviewer for giving time and reviewing our article and we strongly appreciate your feedback that helped us improve our manuscript. Below you will find the answer to each question as well as the information about the changes inserted in the manuscript. All changes are highlighted.

We tried our best to respond to each concern raised.

REVIEWER #3

Sabbatino et al review literature on androgen receptor action in gender-related cancers. This article highlights the role of the AR in prostate, breast, lung, and gastrointestinal cancers. The authors discuss how AR signaling contributes to cancer onset and progression. They also suggest that understanding the role of AR in these tumors could help identify new biomarkers for early diagnosis and develop better therapeutic approaches by directly targeting AR or its downstream signaling. Overall, this paper highlights the importance of studying AR expression in non-classical hormone-dependent cancers to improve cancer treatment options. This review would benefit from more synthetization of the literature to flow better; it is choppy and disjointed in certain sections (as discussed below). Additionally, this manuscript would benefit from further comment/discussion of the potential implications of AR targeting in various disease stages. Lastly, the article has numerous grammatical and spelling errors that need attention. Please consider addressing the following concerns.

We sincerely thank the reviewer for the accurate analysis of our manuscript. Below our point-by-point responses.  

Major Comments:                                   

  • Please comment of what drives expression of different AR isoforms – A vs B. Is their expression gender-related? Disease specific? Organ, tissue, or cell type specific or highly favored? How well does each isoform respond to androgens and anti-androgens?

We ameliorated the paragraph about AR (Pag4, lines123-143, red text market) by adding all the information suggested by the reviewer about the differential expression of the two AR isoforms.

Both the isoforms are expressed in a wide variety of adult and fetal, reproductive and non-reproductive tissues [16]. The full-length AR-B represents the predominant AR species in all tissues in which both isoforms were detected and, in male and female adult reproductive tissues, AR-B is expressed at high concentration, while AR-A comprises 20% or less of the total AR protein [16].  Furthermore, the ratio of AR-A to AR-B did not change widely in the tissues examined [16]. The two AR isoforms slightly differ in their activity and similarly respond to a variety of androgen agonists and antagonists [17]. The unique differences were studied by Liegibel and colleagues [18]. They proved that AR isoforms have distinct functions in human cells of mesenchymal origin such as osteoblastic cells and genital skin fibroblasts. AR-B was responsible for the mitogenic stimulation of mesenchymal cells, while, in AR-positive tissues, AR-A inhibited the mitogenic function of androgen-activated AR-B. The AR-A was unable to to stimulate cell proliferation probably for the reduced binding of AR co-activating protein to the truncated  N-terminal TAD [18].

Both the isoforms are expressed in prostate cancer where AR-B level is still higher than AR-A. Anyway, the AR A/B ratio increases in PCa, in parallel with the Gleason score [15]. These results agree to those obtained in colon cancers where the AR-B expression decreases while the AR-A expression is manteined [19]. The different results in AR A and B activity can be explained considering that all the measurements of their activity are performed using similar levels of the 2 isoforms, but this does not replicate the normal conditions.

  • On line 118, the authors mention how AR mutations play a role in oncogenic signaling. Please comment on key mutations observed to be oncogenic drivers.

We now added more details mentioning some of the hot spot point mutations better studied and able to transform the androgen receptor in an oncogenic driver. Furthermore, we added information about the behavior of some AR splicing variants. Pag. 5, lines 160-172, red text marker.

All the AR domains can undergo mutations. In particular, the selection pressure of drugs on the AR pathway in PCa increases the number of mutations in the ligand-binding domain (LBD) thereby broadening its ligand specificity and sensitivity and reducing clinical treatment effects of PCa and the quality of patient survival. Although various AR mutations have been reported in prostate cancer, specific hot spot mutations (L702H, W742L/C, H875Y, F877L, and T878A/S) were frequently identified after the gain of drug-resistance [23]. A conspicuous group of AR splicing variants, such as AR V1,3,7 and 9, lacks the LBD while showing intact NTD and DBD and, consequently, a constitutive activity [24–26]. Other variants such as AR45 and AR8 lack a DBD and do not work as transcription factors but play different roles. AR8 promotes cell survival via a non-genomic mechanism [27,28]. AR45 is a NTD truncated form, unable to transactivate AR but able to work as a dominant negative and suppress the AR FL functions [28].

  • What other types of cancers have the TMPRSS2-ERG fusions? Are they AR dependent in those as well?

In literature, this fusion is prevalently found in most prostate cancers and indications regarding this fusion in other tumor types lack. Since, however, TMPRSS2, in addition to being a target of AR, is a preferential ETS translocation locus, it can be hypothesized that in other tumors that have deregulations of that gene and respond to androgen treatment, undocumented fusions may be found. Only a manuscript touches upon the presence of this fusion gene in bladder and lung cancers. We modified the text (pages 6-7, lines247-258, red text marker) as follows.

Currently, however, it is not known whether a similar translocation takes place in a subset of lung cancers; indeed, the TMPRSS2–ERG fusion gene seems to be specific to prostate cancer, which may be due to the strong induction of TMPRSS2 by androgen [46]. A series of more than 60000 cancer cases was used to determine the frequency of the TMPRSS2-ERG fusion, assayed by comprehensive genomic profiling (CGP). The fusion gene was detected exclusively in tumor samples from male patients, 30% of which were classified as prostatic cancers. Furthermore, the TMPRSS2-ERG gene was also identified in 4 cases of lung cancers, 4 cases of bladder cancer and 2 of pancreatic cancers [47]. Unfortunately, this seems to be a lonely study demonstrating the presence of the TMPRSS2-ERG fusion gene in non-prostatic cancers. Further analysis might be done to demonstrate if the presence of this AR-related gene can be used in other AR-positive cancers.

  • What is the gender ratio of incidence and survival outcome in lung cancer? Please add to the lung cancer section.

We added the epidemiologic data missed in this paragraph and thank the reviewer for this observation. You can find the following modified data on page 6, lines 209-216, red text marker.

Furthermore, the disease shows sex and gender differences, with a higher incidence in men than women [39] and with sex ratios men: women varying from 1.5 to 2.0. Females tend to be diagnosed younger, at earlier stages and, mostly, with a better prognosis [40]. Worldwide lung cancer mortality is around threefold higher in males with a current downward trend for males and upward trend for females [40,41].  

  • Consider rearrangement of the Renal Cancer section by having an intro paragraph on stats and different types of renal cancer. Then go into the studies depicting positive roles of AR and then negative roles of AR. Mention the gender differences or the gaps that remain in this disease type to further discern the role of AR in renal cancer.

We thank the reviewer. We modified the paragraph according to the suggestion. Pag. 7, lines 279-290, red text marker.

  • What is the prevalence of patients who have PCa treated with anti-androgens and/or AR antagonists that develop bladder cancer? Is this less than patients that never receive any prior AR targeted therapies? What does this suggest about the role of AR in other cancer types beyond prostate and gender-bias ratios observed?

We thank the reviewer for this interesting observation and add the data concerning the relationship between ADT and BCa incidence. We modified the text (pag. 10, lines398-405, red text marker) as follows:

Many studies demonstrate a strong correlation between androgen deprivation therapy (ADT) and BCa incidence. ADT for prostate cancer was associated with a decreased risk of bladder cancer and seems to be a promising therapy for lowering BCa recurrence [85–88]. Other studies fail to identify any impact of ADT on the risk of developing BCa [89,90] but this could be due to several limitations in these observational analyses [90]. For this reason, future methodologically rigorous studies addressing the limitations underlined by Santella and colleagues are needed to evaluate the important potential association between ADT and bladder cancer [90].

  • How do AR targeted therapies directed at dampening the cholesterol biosynthesis pathway impact other hormone nuclear receptors in breast, ovarian, lung cancer, etc? Would this be a beneficial type of treatment for gender-related cancers beyond AR targeting?

Direct targeting of androgen biosynthesis from cholesterol could also impact other steroid hormones production, such as estrogen which is derived from the common synthesis intermediate dehydroepiandrosterone (DHEA). As such, both AR and ER signaling might be impaired using cholesterol biosynthesis inhibitors, thereby affecting different hormone-responsive cancers. However, given such broad influence, the consequences of this kind of therapy might be too unspecific for effective and selective treatment. The effects of this therapy should be evaluated for each single tumor or patients, considering that, when different steroid receptors are simultaneously expressed, they could work in an opposite way as happen in ER positive breast cancer expressing AR (Giovannelli 2018). The use of statins, alone or in combination with other chemotherapeuticseems to be an effective therapy against different types of cancers such as breast, ovarian, pancreatic, lung cancers (https://doi.org/10.1158/1078-0432.CCR-20-1967, https://doi.org/10.1186/s12964-019-0505-5, https://doi.org/10.3389/fonc.2021.595285, https://doi.org/10.12998%2Fwjcc.v9.i18.4500 , doi: 10.1093/jnci/djw275, https://doi.org/10.2147%2FDDDT.S187690, https://doi.org/10.5114%2Faoms%2F123225, https://doi.org/10.1038/s41416-021-01460-4, https://doi.org/10.1016/j.toxrep.2019.10.016, https://doi.org/10.1016/j.tranon.2021.101043. We added these data to the text,  page 16, lines 624-631, red text marker.

  • What is the take-home message? Does gender impact and/or dictate the function of AR in cancers? Or is AR’s role independent from gender and its correlation with incidence and outcome is associated with other contributing factors?

The take-home message of our manuscript is that AR can influence the incidence and progression of gender-related cancer. Its role depends on altered levels of circulating hormones (for pathologies, therapies or excessive administration as happens in transgender persons) and a deep understanding of the mechanisms controlled by AR can be useful not only for developing new therapeutic strategies but also for identifying at-risk subjects and organizing prevention campaigns for lowering cancer incidence. We added this message to the last paragraph “Discussion and concluding remarks”, page 16, lines632-634 and 652-653, red text marker.

  • Please fix grammatical and spelling errors present throughout manuscript. Also make sure one sentence does not constitute a paragraph. Please fix throughout manuscript to combine single statements into complete paragraphs.

We thank the reviewer and apologize for our mistakes. We checked the full manuscript.

Minor Comments:

  • Please define AR when utilizing it for the first time in the text on line 56.

               Thank you for this suggestion. We defined AR on line 56.

  • Prostate Cancer abbreviation is “PCa”. PC is often utilized to abbreviate pancreatic cancer.

We modified in all the manuscript.

  • Define NTD on line 80.

Done.

  • Please keep consistent in utilizing acronyms once they have been defined throughout the text.

We checked all the manuscript. Thank you.

  • On line 142, the authors only reference one ovarian cancer and AR paper, which is a review article. Please make sure to credit the additional studies done on AR in this specific cancer type.

We added 4 new references (refs. 25-31) about AR and ovarian cancer.

  • Make sure to add proper references in text. For instance, on line 165 refer to “Mikkonen and colleague” but reference is not added. Additionally, on lines 173-174 “Recchia et al” reference is not added. Please thoroughly check the entire manuscript and fix the references. This happens in several places.

We accurately checked all the bibliography.

  • Please update cancer statistics throughout manuscript with 2023 stats.

We added the estimated data about incidence and mortality (new cases) in the United States, in 2023. Unfortunately, the most recent Worldwide data are referred to 2020.

  • Make sure to italicize gene names and “in vitro” and “in vivo” terminology throughout the text.

We thank the reviewer for the observation and modified in the text.

  • In Fig 3, please also show that AR interacting with co-regulators leads to alterations in target gene expression. Additionally, SRC kinase also regulates AR and induces a kinase signaling cascade. Consider adding to the schematic. Lastly, add labels of “cytoplasm” and “nucleus” to schematic.
  • We have now modified Fig3.

Round 2

Reviewer 3 Report

Nice edits and additions in response to the revisions. The manuscript is more comprehensive and reads easier.

Author Response

Dear reviewer, we sincerely thank you for your work.
